

# Streamflow Estimation in Ungauged Regions using Machine Learning: Quantifying Uncertainties in Geographic Extrapolation

Manh-Hung Le[1, 2, +], Hyunglok Kim[3,4, +, *], Stephen Adam[5], Hong Xuan Do[6, 7], Peter A. Beling[5], Venkataraman Lakshmi[8]

[1] Hydrological Sciences Laboratory, NASA Goddard Space Flight Center, Greenbelt, MD 20771, USA
[2] Science Applications International Corporation, Greenbelt, MD 20771, USA
[3] USDA Hydrology and Remote Sensing Laboratory, Beltsville, MD 20705, USA
[4] Oak Ridge Institute for Science and Education, Oak Ridge, TN 37830, USA
[5] Virginia Tech National Security Institute, Arlington, VA 22201, USA
[6] Faculty of Environment and Natural Resources, Nong Lam University - Ho Chi Minh City, Ho Chi Minh City, 700000 Vietnam
[7] Center for Technology Business Incubation, Nong Lam University - Ho Chi Minh City, Ho Chi Minh City, 700000 Vietnam
[8] Department of Engineering Systems and Environment, University of Virginia, Charlottesville, VA 22904, USA
*Correspondence to*: Hyunglok Kim (hyunglokkim@gmail.com)
+ former affiliation at Department of Engineering Systems and Environment, University of Virginia, Charlottesville, VA 22904, USA

**Abstract.** The majority of ungauged regions around the world are in protected areas and rivers with non-perennial flow regimes, which are vital to water security and conservation. There is a limited amount of ground data available in such
regions, making it difficult to obtain streamflow information. This study examines how in situ streamflow datasets in data-rich regions can be used to extrapolate streamflow information into regions with poor data availability.. These data-rich regions include North America (987 catchments), South America (813 catchments), and Western Europe (457 catchments). South Africa and Central Asia are defined as data-poor regions. We obtained 81 catchments and 133 catchments for these two data-poor regions, respectively, and assumed they are pseudo ungauged regions for our analysis. We trained machine
learning (ML) algorithms using climate and catchments attributes input variables in data-rich (i.e., source) regions and analyzed the possibility of using these pre-trained ML models to estimate climatological monthly streamflow over data-poor (i.e., target) regions. We found that including diverse climate and catchment attributes in training data sets can greatly improve ML algorithms' performance regardless of significant geographical distance between input datasets. The pre-trained ML models over North America and South America could be used effectively to estimate streamflow over data-poor regions.
This study provides insight into the selection of input datasets and ML algorithms with different sets of hyperparameters for a geographic streamflow extrapolation.

## 1 Introduction

Streamflow is an important variable in fulfilling numerous purposes related to water resources planning and management. For instance, streamflow estimates have been used extensively to represent water availability in many global assessments of



the potential water shortage to meet the increasing water demands in agriculture (Sharpley et al., 2013), forestry (Gush et al., 2002), hydropower generation (Hamlet et al., 2002; De Oliveira et al., 2017), and urbanization (Beighley and Moglen, 2002). Estimates of streamflow have also been used to support global investigation into hotspots of changes in water resources in a warming climate (Veldkamp et al., 2017; Gudmundsson et al., 2021). The most accurate way to measure this variable is to use a stream gauge at any location during a certain period. However, *in situ* streamflow records are usually insufficient to

provide a holistic perspective of water availability in both time and space – owing to the imbalanced distribution of monitoring networks (Do et al., 2020), the quality of archived streamflow data (Hannah et al., 2011), and the declined trends of worldwide available stream gauges (Sun et al., 2018). Also, in many transboundary river basins, records from stream gauges are available but are inaccessible due to political constraints (Gerlak et al., 2011; Kibler et al., 2014). Therefore, characterization of streamflow information at the global scale, especially over ungauged or poorly managed regions, has

been a long-standing challenge of the global hydrology community. The International Association of Hydrological Sciences (IAHS) had identified streamflow prediction in ungauged basins (PUB) as its topic of the decade from 2003 to 2012 (Sivapalan et al., 2003; Hrachowitz, 2013), and has recently reiterated its importance in the so-called "23 unsolved problems in hydrology" (Blöschl et al., 2019).

Prediction in Ungauged Region (PUR) is a new challenge beyond the PUB problem, and PUR has been underrepresented in

the literature. Specifically, The PUR refers to regions that are unmonitored and do not have neighbor representations (Feng et al., 2021), whereas the PUB region still contains some monitoring networks. PUR domains are mainly located in protected areas and rivers characterized by non-perennial flow regimes which are essential to freshwater conservation and water security concerns (Krabbenhoft et al., 2022). PUR was first mentioned by Feng et al. (2021)), who described it as an imbalance between stream gauge settings across continents caused by the fact that in some large regions (e.g., Central Asia

and Africa), few or no single basins are available for study. These sparsely gauged regions challenge PUB techniques as such techniques depend on nearby or similar basins to predict streamflow in ungauged locations. To overcome the PUR problem, Feng et al. (2021) proposed a novel input-selection ensemble, designing different input options and including more widely available data to increase the robustness of their models. However, their experiments only examined PUR-based models within CONUS. Therefore, more studies in different geographic locations are encouraged to provide robust solutions

for PUR.

Recent years have seen rapid growth in the number of hydrologic repositories collecting in-situ streamflow time series to support new advances in PUB and PUR. We now have hundreds of thousands of basins with which to form worldwide hydrologic datasets, providing fundamental platforms for further hydrologic impact studies (Addor et al., 2020). More importantly, these repositories provide various metadata and static basin attributes with consistent formats which can

potentially serve as predictors in streamflow prediction studies (Kratzert et al., 2019b). These repositories include both regional- and national-scale collections in the U.S. (Catchment Attributes and MEteorology for Large-sample Studies, CAMELS; Newman et al. (2015); Addor et al. (2017)); Chile (CAMELS-CL; Alvarez-Garreton et al. (2018)); Brazil (CAMELS-BR; Chagas et al. (2020)) ; the Yellow River in China (China Catchment Attributes and Meteorological dataset,





CCAM; Hao et al. (2021)); the continent of Africa (African Database of Hydrometric Indices, ADHI; Tramblay et al.
(2021)); the continent of Australia (CAMELS-AUS; Fowler et al. (2021)); the continent of Europe (European catchments of
Hydrological Prediction for the Environment model, E-HYPE; Kuentz et al. (2017); LArge-SaMple DAta for Hydrology and
Environmental Sciences for Central Europe, LamaH-CE; Klingler et al. (2021)); the continent of North America (The
Hydrometeorological Sandbox, HYSETS; Arsenault et al. (2020)); and finally the global-scale Global Runoff Database
(GRDB; GRDC) and the Global Streamflow Indices and Metadata Archives (GSIM; Do et al. (2018)). Although these
repositories have not fully addressed the problem of disproportionate distribution of streamflow observations (Gudmundsson
et al., 2018), they provide a wealth of auxiliary information (e.g., catchment attributes or climatology) that could be used to
develop new models to support streamflow estimation.

Among the applications of the established large-sample hydrological data sets, machine learning models are arguably one of
the most promising approaches. The popularity of machine learning (ML) has increased dramatically in hydrologic sciences
(Nearing et al., 2021; Shen et al., 2018; Xu and Liang, 2021). Utilizing large volumes of hydrologic data obtained by remote
sensing, ML can be used to improve flood predictions (Cho et al., 2019) and detect extreme weather events (Racah et al.,
2017; Kim et al., 2019). In rainfall-runoff modeling and streamflow forecasting, regional-based ML models outperform case-
specific based hydrological models in simulating streamflow (Kratzert et al., 2019a). ML model also perform comparably to
distributed hydrological models as emulators which characterize water budgets, and they require less time to run simulations
(Tran et al., 2021). In a PUR study, Feng et al. (2021) reported that ML models could perform more favorable than
traditional hydrologic models in their PUR experiments.

The combination of large-sample hydrologic datasets and ML to better predict streamflow and improve water resources
management has received significant attention in recent years. In fact, the capabilities of long short-term memory (LSTM) in
hydrological simulations have been examined intensively through several large-scale sample modeling studies in CONUS
(Kratzert et al., 2019a; Rasheed et al., 2022), Denmark (Koch and Schneider, 2022), and Great Britain (Lees et al., 2021).
Meanwhile, in France, a large sample of 740 gauged basins was used to investigate the impact of the number of training
basins on regional LSTM-based models (Hashemi et al., 2021). Using the global streamflow index metadata (GSIM) and a
random forest model, Ghiggi et al. (2019) introduced an observation-based global gridded runoff dataset covering the period
from 1902 to 2014. Also using the GSIM dataset, Yang et al. (2019) suggested that a combination of LSTM models and
global hydrologic models could improve the performance of global flood simulations.

While ML algorithms have demonstrated their usefulness in hydrological applications, they often require a large amount of
training data, which is not abundantly available in ungauged areas. Transfer learning is an area of research in the ML field
that addresses using information from one problem, i.e., the source, to improve the performance of a model on another
problem, i.e., the target (Pan and Yang, 2009). Transfer learning is often applied to problems where data collection is limited
in the target problem. In the context of the streamflow-prediction problem, we use well-gauged basins as the source and the
ungauged basins as the target. We test the ability of models trained on data from the source to be transferred to the target
problem.



In this study, we examine a ML-based solution to a real-world PUR problem. Specifically, the climatological monthly streamflow values in data-poor regions (i.e., South Africa and Central Asia) are predicted by pre-trained ML models from

data-rich regions (i.e., North America, South America, and Western Europe). We selected climatological monthly streamflow as the response variable as we could acquire more catchments for our experiments and this variable could also be helpful in ecohydrology and environmental flow assessment (Mcmillan, 2021). To test our hypothesis, we use ML algorithms, as these could be easier to set up than traditional hydrological models. A water resource prediction is vital for ensuring the security of water resources in PUR areas.

## 2 Data

Data preparation in this study consisted of two main steps. First, we selected catchments from the Global Streamflow Index Metadata (GSIM) (Do et al., 2018) based on several constraints: 1) catchment size, landscape information, and time series reliability (please see below for details regarding these constraints). Second, we divided the data sets from the first step into two parts, "source" catchments and "target" catchments, to design the current study.

For the first step above, we obtained streamflow observations from the GSIM's monthly streamflow database. In addition, 14 catchment attributes (see Table 1) were extracted from the GSIM's catchment landscape database. Aerial meteorological characteristics (precipitation and air temperature) for each catchment were estimated using the land surface model-based GLDAS NOAH V2.0 dataset (Rodell et al., 2004). We selected this dataset as it provides global meteo-hydrological variables at 0.25 degree with a long record timeseries (1948-2014). Details of the first step's selection procedure are as

follows:

1.  Spatial data selection: We selected catchments having time series greater than 10 years. We only considered reliable delineated catchments (delineation flags marked high and medium). Catchment areas were constrained to a range from 625 km² to 10,000 km² since these catchments will have multiple pixel coverage from the GLDAS dataset with reasonable areal meteorological estimates (Lakshmi, 2004). Finally, the study period was constrained to the years 1948

through 2014 so that it would overlap with GLDAS NOAH-LSM V2.0's available archive period.

2.  Temporal data cleaning: we removed unreliable streamflow time series values (i.e., monthly values which were negative or were greater than 2,000 cms and monthly values which were estimated using fewer than 15 daily values) (Ghiggi et al., 2019). Finally, we kept catchments that had at least 5 data values per month from 1948-2014.

When the first step was completed, a total of 2,722 catchments had been selected. In the second step, we determined data-

rich regions (the source) and data-poor regions (the target) based on a geographic summary of flow gauge locations by continental region. The largest proportion of global gauges is found in North America, followed by Australia, Europe, and South America (Krabbenhoft et al., 2022). It is noted that stream gauges from Australia do not have catchment delineation flags; therefore, we excluded this region in our regional selection. Siberia, Africa, and Asia are least represented in the global stream gauge network (Krabbenhoft et al., 2022), but we only selected the last two regions because we are interested in their





geographic locations (more sensitivity to global water security). In short, we selected 2,257 catchments from three source regions and 214 catchments from two real-world poorly gauged regions. The three source catchments included North America (S1 region – 987 catchments; 135°W-48°W, 26°N-51°N), South America (S2 region – 813 catchments; 60°W-34°W, 33°S-5°S), and Western Europe (S3 region – 457 catchments; 12°W-30°E, 35°N-60°N). The two target regions included South Africa (T1 region - 81 catchments, 22°E-42°E, 35°S-3°N) and Central Asia (T2 region - 133 catchments,

60°E-121°E, 11°S-40°N). Monthly climatology values (12 months) of streamflow (from GSIM) and meteorological variables (from GLDAS) were aggregated from monthly timesteps. In short, we used a total of 16 predictor variables (2 dynamics- and 14 statics- variables) to develop ML models (Table 1). Please refer to **Section 3** below for more detail regarding this methodology.

   Figure 1 shows the distribution of the final selected catchment locations. The files "GSIM_designed_PUR.xlsx" provided in

the Supplement lists this subset of GSIM stations in this study, while Figure S1 in the Supplement shows the locations of catchments from original GSIM, after the first step selection, and after the second step selection.

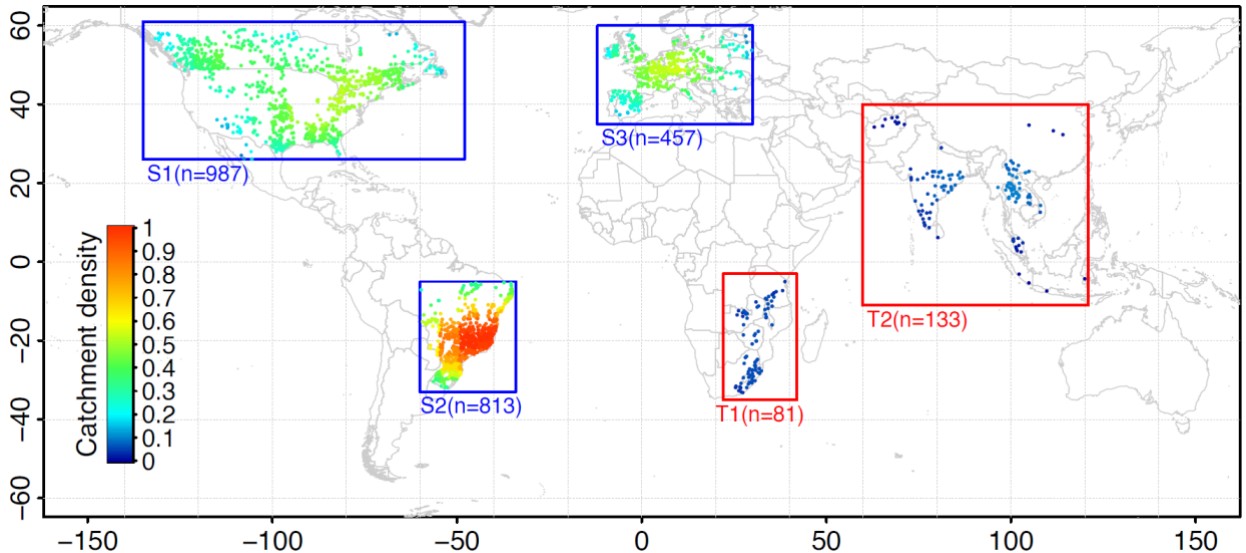

**Figure 1. Distribution of catchments used in this study. Blue circles are source catchments at three regions S1 (North America), S2 (South America), and S3 (Western Europe). Red circles are target catchments at two regions T1 (South Africa), and T2 (Central**
**Asia).**





**Table 1. List of machine learning model predictors**

| No | Name | Description | Sources |
|---|---|---|---|
| 1 | P | precipitation | GLDAS |
| 2 | T | air temperature | GLDAS |
| 3 | long | longitude | GSIM |
| 4 | lat | latitude | GSIM |
| 5 | area.meta | catchment size information from metadata in km$^2$ | GSIM |
| 6 | altitude.dem | height of gauge above sea level (m) reported from dem | GSIM |
| 7 | dr.mean | average catchment drainage density | GSIM |
| 8 | ele.mean | average values of catchment elevation | GSIM |
| 9 | ir.mean | average of catchment irrigation area | GSIM |
| 10 | slp.mean | average of slope within catchment boundary | GSIM |
| 11 | pd.mean | average values of population density within catchment boundary (GPWv4 - 2010) | GSIM |
| 12 | sb.mean | average values of bulk density in kg/cubic meter within catchment boundary | GSIM |
| 13 | scl.mean | average values of clay content in soil profile within catchment boundary | GSIM |
| 14 | snd.mean | average values of sand content in soil profile within catchment boundary | GSIM |
| 15 | slt.mean | average values of silt content in soil profile within catchment boundary | GSIM |
| 16 | tp.mean | average values of topographic index within catchment boundary | GSIM |

## 3 Methodology

### 3.1 Machine learning (ML) models

For this study, we selected three widely-used machine learning algorithms: support vector machines, random forests, and gradient boosted trees. Though these models are commonly used for classification tasks, each of these models can be applied to a regression problem, as required for predicting streamflow.

### 3.1.1 Support Vector Machines (SVMs)

The SVM algorithm, which was first introduced in the late 1970s (Noble, 2006), is based on a supervised non-parametric
statistical learning technique that separates data into classes by optimizing separation hyperplanes. The strength of SVMs lies in its ability to solve the convex quadratic optimization problem to obtain a globally optimal solution (Pisner and Schnyer, 2020). It has been demonstrated that SVMs require a smaller training set to produce higher accuracy than a traditional classifier (Mountrakis et al., 2011; Tamiminia et al., 2020). The hyperparameters for SVM were tuned by trial and error for the C-parameters (controlling the boundary of the hyperplane) and kernel functions (transforming input data).





Please see Table 2 for more details on hyper-parameters and their ranges in finding minimum of the loss objective function during the model training.

### 3.1.2 Random Forest (RF)

The Random Forest (RF) is a tree-based ML algorithm (Segal, 2004) used in a number of regression studies as a robustness technique (Booker and Woods, 2014; Desai and Ouarda, 2021). RF is based on the decision tree method with an ensemble

learning approach, utilizing boosting and bagging procedures to solve the regression problem. The boosting technique integrates multiple models to solve the same problem, thus increasing the prediction accuracy. The accuracy of RF regression relies on two RF's hyperparameters, the number of trees and the number of features randomly sampled for each tree in the forest. The regression accuracy of RF is more sensitive to the number of features sampled for each tree than the number of trees, where fewer features would shorten the processing time but result in less accuracy (Kang and Kanniah,

2022). It is common to set the number of trees as high as possible since doing so will not reduce efficiency or cause overfitting during the regression process (Belgiu and Drăguţ, 2016). Please see Table 2 for more details on hyper-parameters and their ranges in finding minimum of the loss objective function during the model training.

### 3.1.3 eXtreme Gradient Boosting (XGBoost; XGB)

XGB is a scalable ML technique for tree boosting (Chen and Guestrin, 2016). The algorithm utilizes Classification and

Regression Trees as base classifiers and is jointly decided by multiple related decision trees whereby the input sample of each decision tree relates to the training and prediction results of the previous decision tree. XGB is a highly flexible and versatile tool that can solve most regression and classification problems. In this study, we tuned the XGB model based on the learning rate (η) and gamma (γ). The learning rate determines the step size of each iteration as the XGB model optimizes its objective, while γ is a pseudo-regulation parameter used to prune the model. Please see Table 2 for more details on hyper-

parameters and their ranges in finding minimum of the loss objective function during the model training.



**Table 2. List of hyper-parameters used for each machine learning algorithm in this study**

| Algorithms* | Hyper-parameters** | Value ranges/ options |
|---|---|---|
| **Support Vector Machine (SVM)** | C-type (type) | eps-regression; nu-regression |
| | kernel functions (kernel) | linear, polynomial, radial, sigmoid |
| **Random Forest (RF)** | Number of tree (ntree) | [2,50], step 2 |
| | Number of features (mtry) | [1,16], step 1 |
| **XGBoost (XGB)** | Learning rate ($\eta$) | [0.1, 0.9], step 0.1 |
| | Gamma ($\gamma$) | [0.1, 15], step 0.2 |

Note:

*All three algorithms were employed in R programming language. We used 'svm/e1071', 'randomForest/randomForest', and 'xgboost/xgboost' as functions/packages to implement SVM, RF, and XGB, respectively.

**variables in parenthesis are parameters corresponding to the hyper-parameters for each function used in the R programming language.

## 3.2 Experimental design

We aimed to examine whether ML models could be trained on one region (the source), transferred to another region (the target), and accurately predict climatological streamflow without training data from the target. We investigated the model performance and the ability to transfer models for each of the three ML algorithms (SVM, RF, and XGB) using the experimental design outlined in Table 3. The following paragraph describes procedures of three ML algorithms (SVM, RF, and XGB) in predicting climatological monthly streamflow at local-based models (using target catchments to train the ML algorithms) and source-based models (using source catchments to train the ML algorithms).

The development of target streamflow prediction (two target regions) using local-based models was as follows:

1.  Each training (25% of the total number of data), validation (25%), and testing (50%) dataset was re-scaled to [0.1, 0.9] using the minimum/maximum scaler. Data in the training set is rescaled in using the scaled range [0.1, 0.9] to accommodate extreme values that could be present in the validation and test sets. A logarithm was used to transform the response variable (climatological streamflow) as we found that the distribution of the streamflow data set's logarithm likely followed a normal distribution (please see Figure S2 in the Supplementary materials for more details).

2.  We trained three ML algorithms with the training- validation dataset by adjusting the hyper-parameters of each algorithm given in Table 2 and validated these trained ML algorithms with the testing dataset. Mean Square Error is the objective function for all algorithms.





3.   The modified Kling-Gupta Efficiency (KGE, Gupta et al. (2009)) method was used to evaluate streamflow prediction. Numerous studies consider the KGE score to be an acceptable metric for quantifying the performance of hydrologic model simulations (Le et al., 2022; Lin et al., 2019).

$$KGE = 1 - \sqrt{(r-1)^2 + (\beta - 1)^2 + (\gamma - 1)^2} \qquad (1)$$

In which:

$r$ is the Pearson correlation coefficient, reflecting the error in shape and timing between observed and simulated

streamflow.

$\beta$ is the bias term, evaluating the bias between observed and simulated streamflow.

$\gamma$ is the ratio between coefficients of variation in observed and simulated streamflow, assessing the flow variability error with bias consideration. KGE value of the mean flow benchmark is -0.41 (Knoben et al., 2019).

   4.   We repeated steps (1) through (3) for 100 training, validation, and testing datasets so that we obtained 100 prediction

results for each ML algorithm. From these results, we calculated the median and 95[th] and 5[th] percentile of the KGE scores. Using the range of these scores, we could evaluate the robustness of different ML algorithms over the testing datasets.

An assessment of target streamflow prediction using source-based models (with seven experiments) proceeded as follows:

-   We implemented steps (1) through (4) again to create 100 models corresponding to 100 different input sets.

-   We used the 100 predicted models for each source-based ML algorithm to predict climatological streamflow in 100 different testing data sets at target regions. In total, 10,000 prediction results (100 ×100) were obtained, and the median and 95[th] and 5[th] percentile of KGE scores were calculated accordingly.



**Figure 2. Flow chart of this study**






**Table 3. List of experiments used in this study.**

| Experiment | S1 | S2 | S3 |
|---|---|---|---|
| EX1 | X | | |
| EX2 | | X | |
| EX3 | | | X |
| EX4 | X | X | |
| EX5 | X | | X |
| EX6 | | X | X |
| EX7 | X | X | X |

Note: S1, S2 and S3 represent North America, South America, and Western Europe, respectively.

## 4. Results and Discussion

### 4.1 Assessment of the performance of local-based and source-based ML algorithms

This section describes the performance of the three ML algorithms (SVM, RF, and XGB) in predicting climatological monthly streamflow at local-based models (using target catchments to train the ML algorithms) and source-based models (using source catchments to train the ML algorithms). The local-based models also served as benchmark models.

   In Figure 3, violin plots are used to visualize the performances of three ML algorithms on EX7's testing set. We only present EX7 as we observed almost similar simulated behaviours from other experiments. For each algorithm, twelve models
(corresponding to the twelve months of the year) were developed to predict climatological streamflow. In each model, we used 100 different training-cross-validation-testing data sets to perform 100 different simulations. The median point, the length of grey bar and the shape of violin plot indicate that the ML algorithms behaved differently when trained by different datasets; these results are beneficial in evaluating the algorithms' uncertainty.

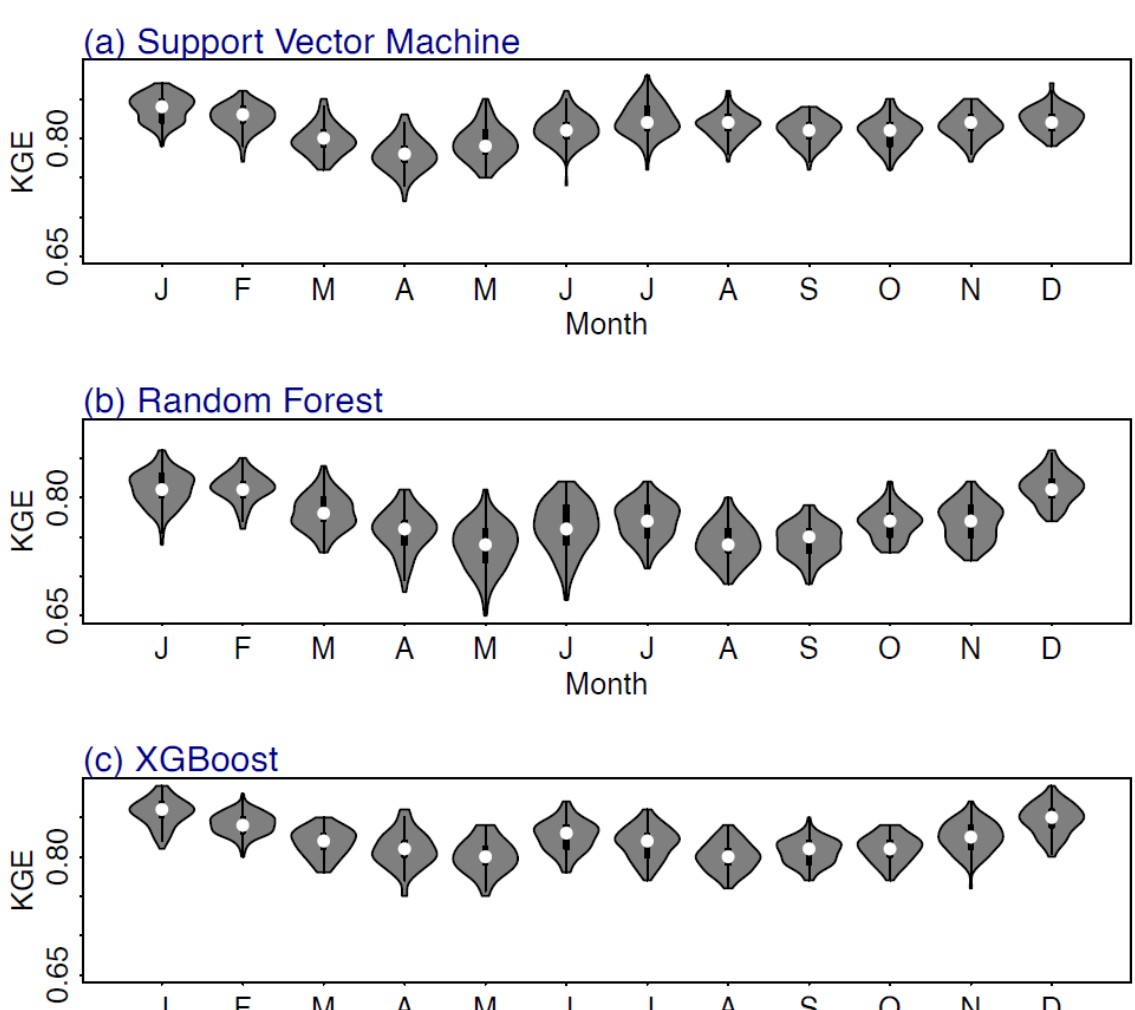

**Figure 3. Illustration of 100 simulations for (a) Support Vector Machine (b) Random Forest, and (c) XGBoost in EX7 using violin plot. The white dot represents the median. The thick gray bar and thin gray bar in the center denote the interquartile range and the rest of the distribution, respectively. The violin plot's wider sections indicate a higher distribution of KGE values.**

Figure 4 derived from the two benchmarks and seven experiments employing different training samples, demonstrates the relationships between the KGE scores of sample sizes and climatological streamflow on testing sets. The figure shows relationships found in each of the four seasons (Winter-DJF, Spring-MAM, Summer-JJA, and Autumn-SON). Although all ML algorithms exhibited statistically significant correlation between sample size and performance as test sets, the tree-based ML algorithms seem to be more pronounced, with a higher correlation coefficient found with RF and XGB than with SVM. Inter-seasonal comparison shows that the accuracy of climatological streamflow prediction during winter months (Figure 4 a-1, b-1, c-1) seems to be associated with sample size. In seasons other than winter, SVM seems to be less sensitive to sample size, especially when the total sample size is over 200, as indicated by KGE scores which almost always varied by



sample size (Figure 4 a-2, a-3, a-4). In short, ML algorithms performed better with more training datasets, even though their distributions vary greatly when input datasets from different sources are combined (see Figures 5-6-7). This finding indicates the importance of considering different climatic regions for transfer learning or PUR problems in hydrology. Table S1 and Table S2 in the Supplementary materials provide numerical KGE scores in each month for two benchmark models and seven

experiments, respectively.

**Figure 4. Relationship between total training samples and performances at testing dataset using (a) SVM, (b) RF and (c) XGB with different input setting. This assessment is at four different seasons (1) winter (DJF), (2) spring (MAM), (3) summer (JJA), and (4) autumn (SON). Bottom right values in each panel indicate Pearson correlation coefficient between sample sizes and KGE scores.**






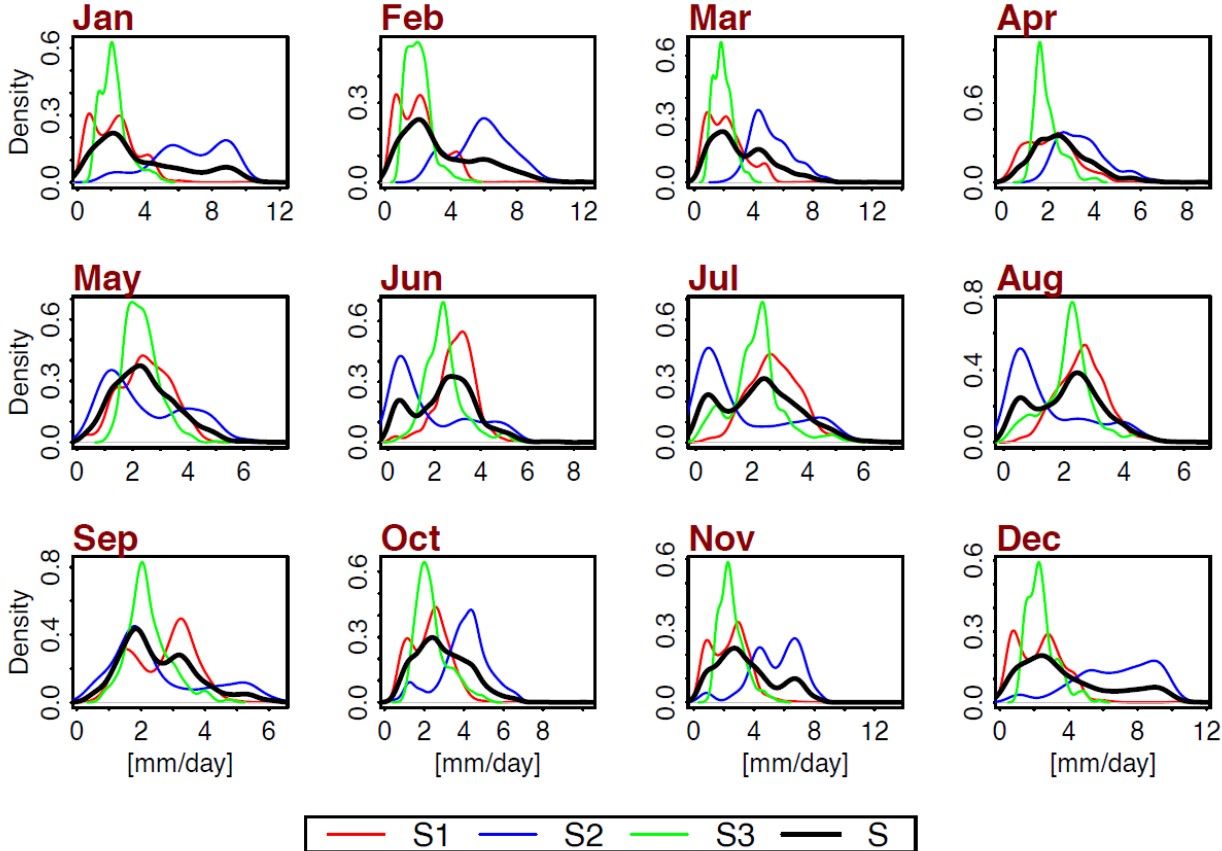

Figure 5. The density plots of climatological precipitation from different input datasets, including S1 (North America) (red lines), S2 (South America) (blue lines), S3 (Western Europe) (green lines), and S (S1+S2+S3) (black lines)





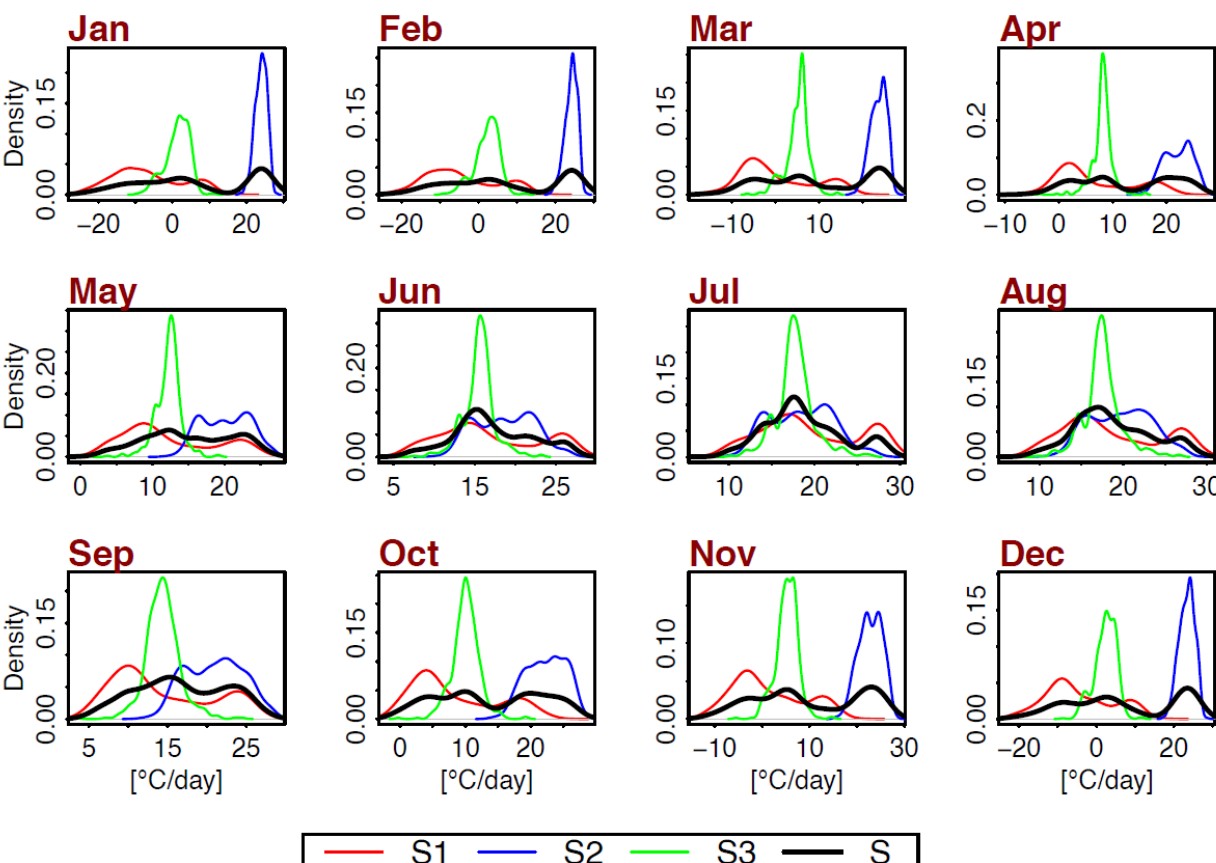

Figure 6. Similar to Figure 5 but for climatological air temperature.





**Figure 7. Similar to the Figure 5 but for fourteen catchment attributes.**

## 4.2 Performances of source-based ML algorithms at target regions

This section evaluates the uncertainties found when we employed source-based ML algorithms to predict climatological
streamflow at target regions without retraining with data from the target region.

Figure 8 shows the performance of the models trained using data from the source regions used to target climatological
streamflow prediction across the T1 region. We used local-based SVM algorithms for our benchmark models as they showed





the best overall performance of the various local-based ML algorithms (see Section 4.1). Generally, source-based ML algorithms were less successful than these benchmark models in predicting climatological streamflow. Specifically, averaged

differences in KGE scores between the source-based ML algorithms and the benchmark models were negative across all algorithms and input settings. This result is understandable as source-based ML algorithms were trained with no prior input-output information of the target regions. On the other hand, the benchmark model was developed based on information of the input-output relationship at the target regions, although the total number of trained samples in the benchmark was small. With regard to the inter-comparison of ML algorithms, the source-based SVM algorithm seems to produce higher

uncertainty with a wider range of KGE scores than the two other tree-based algorithms, especially with regard to input data sets from EX2 (Figure 8 b-1), EX3 (Figure 8 c-1), and EX6 (Figure 8 f-1). When we examined different experimental input settings, source-based ML algorithms forced by the EX1 dataset seem to provide the most reasonable KGE scores, as they have fewer negative values than the benchmark models under all three ML algorithms (Figures 8 a-1, 8 a-2, and 8 a-3). Interestingly, a higher number of sample sizes in the source-based models did not mean that these source-based models

performed well at the target region when predicting climatological streamflow. For example, source-based models created using EX7 (formed by combining all input sets from the three sources regions) produced typically low KGEs scores compared to other input settings (Figures 8 g-1, 8 g-2, and 8 g-3).



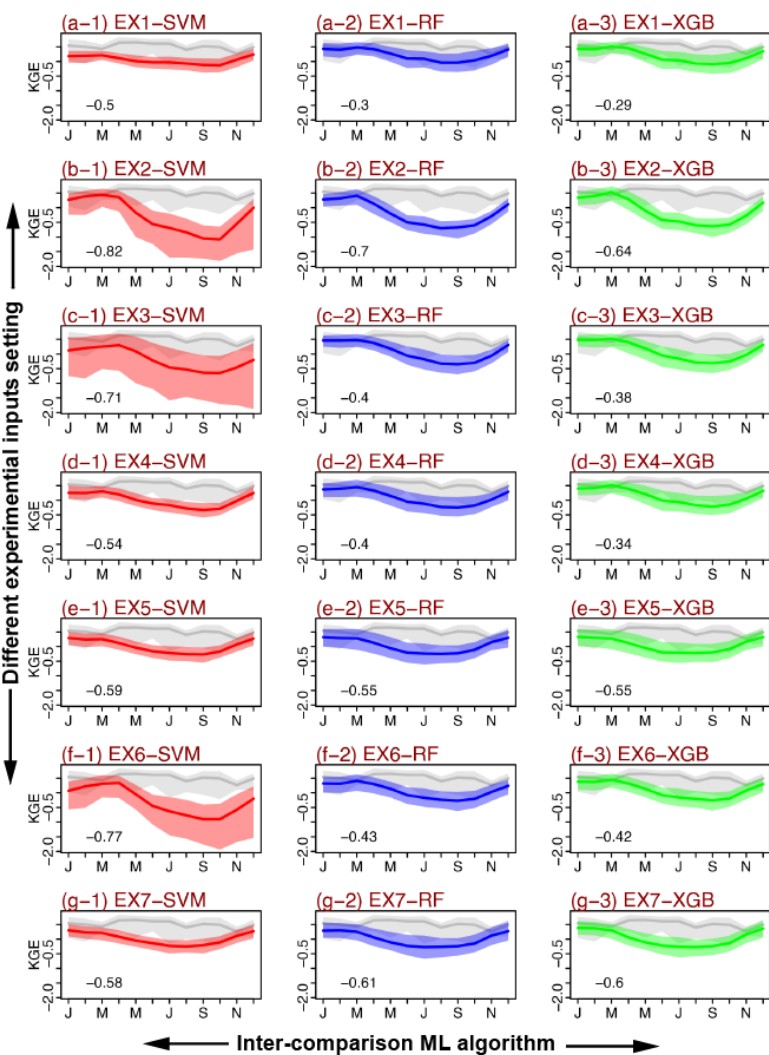

**Figure 8. Comparison climatological streamflow prediction performances between different source-based ML algorithms (1) SVM, shaded red area (2) RF, shaded blue area (3) XGB, shaded green area as forcing by different input settings EX1-EX7, (a)-(g) at T1 region (South Africa). Local-based SVM algorithm was selected as benchmark model (shaded grey area). Bold lines (grey, red, blue, and green) in each panel indicate median KGE scores. Bottom left values exhibit on average difference between source-based ML algorithm and benchmark model across 12 months**

Figure 9 recreates the exercise demonstrated in Figure 8, this time using the T2 region (Central Asia) as target. Again, we observed negative values between the different source-based ML algorithms and the benchmark models (local-based SVM algorithms) across experimental scenarios. The uncertainties of the source-SVM algorithm were also high when these models were used to predict climatological streamflow datasets in the T2 region, especially with EX3 and EX6 (Figure 9 c-1 and Figure 9 f-1). Results were mixed with regard to different combinations of input sets and ML algorithms. Specifically, with the SVM algorithm, the source-based models using EX2 input (Figure 9 b-1) produced KGE scores closest to the benchmark model, with an average difference of -0.1. On the other hand, source-based RF and XGB showed the best





performance with the target region's data sets when EX1 was used as the input data set (Figures 9 a-2 and 9 a-3). We also determined that there was no significant difference when we brought source-based models into the EX7 dataset (forced by all three sources datasets) to predict climatological streamflow in the T2 region as compared to the other experimental settings.

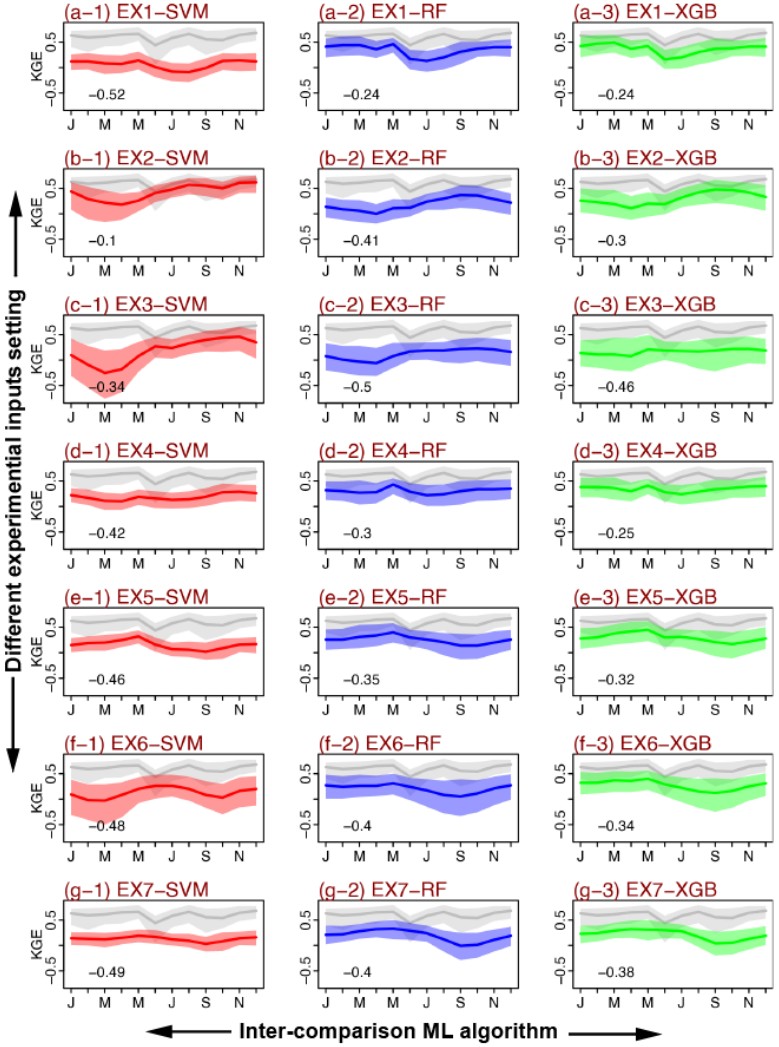


**Figure 9. Similar to Figure 5 but for T2 region (Central Asia).**

Figure 10 shows the relationships of seven experiments performed at the source and corresponding climatological streamflow prediction performance of these experiments at the target across all four seasons. Here, we observed an insignificant relationship between the KGE scores at the source versus the KGE scores at the target. In inter-comparison of seasonal performance for the T2 region, source-based ML algorithms' climatological streamflow predictions at target seem
consistent across seasons. On the other hand, for the T1 region, KGE scores at target were typically low during summer



(JJA) and autumn (SON). In short, we expect the source-based model to produce more promising results at T2 (Asian watersheds) than at T1(African watersheds).

**Figure 10. Scatter plot between streamflow prediction performances at sources and streamflow prediction performances at target in four different seasons (1) Winter, DJF, (2) Spring, MAM, (3) Summer, JJA, and (4) Autumn, SON using three ML algorithms (a) SVM (b) RF and (c) XGB. Red and blue symbols were performances of T1 (Africa) and T2 (Asia) region, respectively. Horizontal dash line is a threshold that if KGE scores fall behind it, the observed mean is a better predictor than the model.**

## 5 Limitations and further studies

In this study, the quality of precipitation and air temperature derived from GLDAS was assumed to be the same around the world. The GLDAS, however, may have better quality in regions with more ground data. While this study focuses on climatological streamflow prediction, finer timesteps of streamflow (e.g., daily or monthly) could be investigated in the future. As an alternative to the three ML algorithms used in this study, additional ML algorithms may be considered for testing with transfer learning problems. There are also promising topics regarding input selection based on climate regions and error prediction in different climate regions.





## 6 Conclusions


An attempt is made in this study to understand the uncertainties associated with the Prediction Ungauged Region (PUR) topic in a real-world setting. By using ML algorithms, we investigated how well a pre-trained model from the source (rich-data regions) could be employed to estimate streamflow in the target region (data-poor regions). Instead of developing a novel ML algorithm, we use ML algorithms to quickly test our hypothesis since ML algorithms could be easier to set up than

traditional hydrological models. Specifically, we employed three ML algorithms (Support Vector Machine, Random Forest, and XGBoost) in predicting climatological streamflow in two real-world sparsely-gauge regions. The two sparsely gauged regions, South Africa and Central Asia, were defined as target regions. We performed seven experiments combining three data-rich regions (defined as source regions): North America, South America, and Western Europe. We developed source-based models based on different input settings and different ML algorithms, then used these models to predict climatological

streamflow at the two target regions. Our study produced three major findings:

- By experimenting with different input combinations, different climates and catchments attributes do not introduce noise into ML algorithms, but rather enhance their performances.
- We must carefully select the source data sets used to train source-based ML algorithms as experiments with more sample data from sources did not guarantee that our source-based models would perform better in predicting streamflow

at the targets. Two sources which can be used as potential data sets in creating source-base ML algorithms are North America and South America.
- With regard to inter-comparison of the three ML algorithms used in this study, XGB seems to be the most robust algorithm as it (i) performed well when trained with local datasets and (ii) predicted streamflow at target regions with a narrower range of uncertainties than other ML algorithms.

Most PUR areas are concerned with water security and predicting water resources in these areas is crucial. In this study, ML techniques are demonstrated to be capable of solving hydrological problems in these regions..

## Acknowledgment

We sincerely acknowledge the support of the Army Research Office (Proposal No. 78464-EG "Connection between flash drought and water security using AI/ML algorithms for certain African and Asian Watersheds"), program manager Dr. Julia

Barzyk. The GSIM data sets were obtained at https://doi.pangaea.de/10.1594/PANGAEA.887477 and https://doi.pangaea.de/10.1594/PANGAEA.887470, and the GLDAS data sets were obtained at https://disc.gsfc.nasa.gov/. These datasets can be accessed at https://osf.io/9xsuj/ (10.17605/OSF.IO/9XSUJ). Code is available at https://github.com/mhle510/PUR_experiments.





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
