# Peer review of "Streamflow Estimation in Ungauged Regions using Machine Learning: Quantifying Uncertainties in Geographic Extrapolation"

_Hydrology and Earth System Sciences, 2022_

## Community Comment (CC1)

```python
import umap
mapper = umap.UMAP(n_neighbors=20).fit(overall_post)
import umap.plot
umap.plot.points(mapper, labels = labels)
```

 <AxesSubplot:>

---

## Community Comment (CC2)

[Figure]

Figure 1. Uniform Manifold Approximation and Projection (UMAP) clustering of the source and target regions that are colored and symbolized by each catchment (Source Region 1 (S1)-North America, Source Region 2 (S2)-South America, Source Region 3 (S3)-Europe, Target Region 1 (T1)-South Africa, Target Region 2 (T2)-Central Asia)

---

## Author Comment (AC1)

**Response to RC#1**

Title: Streamflow Estimation in Ungauged Regions using Machine Learning: Quantifying Uncertainties in Geographic Extrapolation

General:

[1] This paper attempts to make predictions of monthly averaged streamflow in data scarce regions with machine learning models that were trained in data rich regions. They test their predictions with different permutations of training regions. As expected, the models perform better with different climates and catchments attributes in the training set. Interestingly, however, thew results suggest that models trained in North and South America are more reliable than models trained in Europe. They also find, as expected, that extreme gradient boost outperforms support vector machine and random forest. The paper is written fairly well, with exceptions noted below, and provides additional support for the well established conclusion that machine learning models trained on diverse data sets can be useful outside the basins which they are trained. This paper expands that conclusion by transferring the learned models to entirely new regions, in particular to data sparse regions, which is important, as the authors point out.

Response: Thank you for your positive comments and valuable suggestions which will help us improv our paper's quality. Below we have answered your concerns point by point.

[2] It was not clear to me if these models were forward looking or backward. I am not entirely sure how useful a monthly average streamflow prediction is in practice, especially if the forcings which drive the prediction are aggregated over that particular month, which would have the prediction a backward estimate. If, however, the forcings are aggregates from the previous month, then this is valuable to water resources management. I ask the authors to make this clarification in their data and methodology sections.

Response: Our current analysis is backward looking. We understand the limitation you mention and will address your concern in the revised manuscript. That being said, our study is unique in that few attempts have been made to examine the transfer learning concept in a real-world context in the field of hydrology. We tested the pre-trained ML models' streamflow prediction using the assumption of no prior knowledge of the new streamflow prediction domains. These new streamflow prediction domains are located in data-poor regions (Central Asia and South Africa) where water security is a long-term problem. In reviewing the case study, we saw that backward analysis would be particularly useful to us. From the GSIM datasets, aggregated climatological streamflow is the most feasible way to obtain more sample points for our analysis. More importantly, in transboundary river territories where obtaining in-situ streamflow can be challenging, the climatological streamflow is beneficial for long-term water resources planning. These averaged climatological streamflow indices are considered to be the minimum number required for water resource planning in a transboundary river basin in Northeast Vietnam (NAWAPI, 2018).

References:

NAWAPI, 2018. Bang Giang - Ky Cung Water Resources Planning Project, Water Resources Assessment Report (in Vietnamese), Ministry of Natural Resources and Environment, Hanoi.

[3] This paper omits non-machine learning models from the study because they are harder to set up. And unfortunately, there is no benchmark model/s presented. I believe that this could potentially draw criticism. I do fully understand the need for easy-to-use models in some situations. I would encourage the authors to

rethink their framing of the model selection in the introduction and conclusion. Perhaps it would be good to make a case for the benefits of easy-to-use models, and make a case that these shallow learning models are suitable for monthly averaged streamflow over the state-of-the-art LSTM mode, which has been shown again and again to outperform other streamflow models, even when trained out of sample.

Response:

We thank the reviewer for their suggestions. In the revision, we plan to compare our reported ML models with an state-of-the-art land surface model (i.e., Noah-MP4.0.1) that uses four different routing schemes: "free drainage", "groundwater", "TOPMODEL" and "BATS". In addition, we will also consider developing LSTM and GRU deep learning models to assess whether they will introduce substantially different outcomes.

Abstract:

[4] Line 21 has double periods.

Response: Thank you for noting this typo. We will revise accordingly.

Introduction:

[5] Lines 38 and 39 claim about stream gauges being the most accurate way to measure streamflow is vague and trivial. Are you making a distinction between remote sensing and in situ measurements? There are many methods of gauging a stream, some more accurate than others. I'm not sure what is the purpose of the sentence, remove or clarify.

Response: Thank you for this note. We will revise accordingly.

[6] Lines 78 and 79: If there is a good argument that ML is not **the** most promising approach, I'd like to see a citation. Otherwise just state it directly as "machine learning models are arguably one of the most promising approaches"

Response: Thank your for your valuable suggestions. We will revise accordingly.

[7] Line 86: I'm not sure it is obvious what at "traditional" hydrological model is.

Response: by "traditional" hydrological model, we mean a physical-based hydrological model such as SAC-SMA, VIC, mHM, and the National Water Model. We will revise these lines accordingly.

[8] Line 107-108: I assumed your hypothesis was about ML model's ability to transfer learning from one region to another, but here you claim that you use ML models because they are easier to set up?

Response: Thank you for your interesting comments. By "easier to set up," we were comparing our ML models with physical-based hydrological models. Setting up a physical-based model for individual catchments requires considerable processing time and assumptions of soil type, topography, and physics. Such a model is challenging to work with in handling large pool catchments. We will rephrase this sentence to clarify it for readers.

[9] Lines 108-109: I think the last sentence of this paragraph is fragmented. What kind of water resources prediction? In what context are the water resources secure or insecure?

Response: Our results are helpful for long-term water resource planning as our model outcomes (climatological streamflow) can provide general information for water resource managers in regions with limited or no access to in-situ networks. This information is probably essential in many transboundary rivers. We will revise this paragraph for the sake of clarification.

Data:

[10] Line 127: What is the rational for removing values greater than 2,000 cms?

Response: Monthly streamflow values greater than 2000 cms do not seem feasible, so we excluded them to avoid noisy data in the post-processing datasets (Ghiggi et al., 2019).

References:

Ghiggi, G., Humphrey, V., Seneviratne, S. I., and Gudmundsson, L.: GRUN: an observation-based global gridded runoff dataset from 1902 to 2014, Earth System Science Data, 11, 1655-1674, 2019.

[11] Line 140: Can you make it clear if your model is making a forward or backward prediction? Is your monthly forcing aggregates from the same month in which your monthly averaged streamflow comes from?

Response: In its current state, this work is a backward prediction forcing of aggregated values from the same month for P and T.

[12] Figure 1: What unit is catchment density?

Response: It is a non-dimensional unit reflecting the relative degree of dense catchments over a specific region.

Methodology:

[13] Lines 207-209: This wording is a little confusing. Can you rephrase to make it clear that the validation set was used to tune the hyper-parameters? Meaning, your training set is used to get the model weights, and then you check the quality of those weights by calculating an error on the validation set, then modify a hyper-parameter and train again, then check the quality of the new weights on the validation set. And to be clear, you do not calculate any error on the test set until the hyper-parameters have been chosen, right?

Response: Your understanding is correct. We will revise accordingly.

[14] Table 3: Consider moving the regions into the table header, instead of as a note.

Response: Thank you for this note. We will revise accordingly.

**Results and Discussion:**

[15] Line 236: "The local-based models also served as benchmark models" This should be moved to the methods section.

Response: Thank you for this note. We will revise accordingly.

**Limitations and further studies:**

[16] Line 326: In parentheses you have "daily or monthly", but I think you meant "daily or hourly"

Response: Thank you for this note. We will revise accordingly.

Conclusions

[17] Line 334-335: "ML algorithms to quickly test our hypothesis since ML algorithms could be easier to set up than traditional hydrological models." I think this is a bad reason to us ML. There is no use doing a study with one tool instead of another simply because it is easier.

Response: We greatly appreciate your point. In our experimental setting, we would use a straightforward term to describe the advantage of ML over traditional hydrological models [or physical-based models]. We will revise our word choice to better explain our model selection.

[18] Line 351: double periods.

Response: Thank you for this note. We will revise accordingly.

---

## Author Comment (AC2)

**Response to RC#2**

[1] This paper investigated the monthly streamflow prediction in ungauged regions using Machine Learning (ML) based methods. The authors compared three ML methods in global basins with two large regions as data-poor targets. The overall structure of this ms is clear to follow and the topic is intriguing to me. I have some comments as shown below on better clarifying the methodology and performing more profound discussions on the results to safely draw the conclusion. Hopefully these suggestions can help to improve the quality of this study.

Thank you for your positive comments. We have answered your questions and comments point by point below. We will incorporate your suggestions to improve our manuscript's clarity.

**[2] Introduction**: The authors did a good job here with a comprehensive review on the present studies and I enjoyed reading this part.

Thank you for your kind comments.

**Methodology:**

[3] To my knowledge, the present cutting-edge ML applications in streamflow prediction mainly focus on daily prediction with deep learning models like LSTM which show superior performance over other models as shown by several studies already cited in this ms. The advantage of DL models over traditional ML was not only shown in hydrology but also in many other fields. I feel the authors may discuss more on the motivation of their choices on monthly prediction and model selection with traditional ML methods.

This study focuses on data-centric ML rather than method-centric ML. Our significant contribution is proposing a new dataset for testing Prediction in Ungauged Regions (PUR) in a real-world case study. Our study is unique in that we have attempted to predict streamflow across continents with diverse climatic and catchment characteristics. Previous works with LSTM often used datasets which required minimum effort to process and which focused on a single continent. Since data availability in our experiments varies from place to place, we selected climatological monthly predictions to maximize the number of available catchments. Our focus is intended to solve a spatial prediction (predicting streamflow in different geographic/ continent regions). At the same time, LSTM is favored for temporal prediction; thus we did not consider LSTM in this manuscript. Finally, using this method, our sample data for the training set could consist of fewer datasets than would normally be favorable for LSTM.

[4] Better clarification on the framework and experiment design is needed to help readers easily understand the method section. I am quite confused about the meaning of "100" mentioned in line 219 and throughout the ms. Does this mean a 100-fold cross validation to cover all the available data? If so, there would be no basin overlapping for each testing but how the 100-simulation range comes then? I also didn't understand how the training, validation and testing dataset were formed with limited details given.

Your understanding is correct. The 100-fold cross-validation was used to assess the uncertainty of each ML algorithm. For one input set, we randomly selected samples with replacements. We repeated this process 100 times, so one catchment could be in the testing set in one fold but in the training set or cross-validation set in another fold. We did not fix the testing set since we wanted to examine the sensitivity of the ML algorithms with different spatial data combinations.

[5] How do you organize and divide data in the time dimension? The streamflow prediction is a time dynamic problem and I see the authors use data across multiple decades, however I only find the results reported for 12 individual months without time continuous information given.

We aggregated monthly data to climatological monthly data. This is why you see our results reported for 12 individual months rather than continuously. As we mentioned previously, our rationale here was to observe streamflow data availability; climatological streamflow is the best way to obtain the maximum number of catchments.

[6] If I understand correctly, the authors train individual models for different months. I am just curious how this choice was made and how the model would behave with one model trained on data from all months instead, especially given the power of ML models handling big data.

Since different months have different seasonality cycles, we believe that having a separate model for each month would be better than having one model for all months. We did test one model for all months, but we saw that this model did not perform well as the separate models for each month.

**Results:**

[7] Reading through the result section, I hope the authors can do a more profound analysis and discussion on their results, not limited to simply describing the figures. The present figures are kind of redundant to me especially without many discussions involved. You may consider removing some unnecessary ones.

Response: We appreciate this comment. In the revised manuscript, we will enhance our results section by incorporating your suggestions.

[8] For the PUR performance evaluation, the authors need to clarify more about the absolute performance in target regions, not only the performance difference from the local models. It's quite intuitive to get worse PUR performance compared with local models, but the readers care more about the direct evaluation, like how will ML models behave and can we get functional models for predictions in ungauged regions? Looking at Figure 8, I feel the absolute PUR performances are mostly close to KGE value of 0.0 (y axis starting at -2.0 can be somehow misleading to readers), which implies unsatisfactory performance for a functional model.

Response: Thank you for your interesting note. We will add a direct values comparison to our revised manuscript. We observed positive PUR performance (KGE >0) over the course of several months. We used -2.0 to show the full possible ranges of KGE performances from the source-based models demonstrating model uncertainties in predicting streamflow at the target region. In the revised manuscript, we will provide a more detailed explanation to support our assertion that functional models can perform quite well.

[9] It's quite interesting and also surprising to me for the statement of line 290 that including more training data (EX7 here) leads to lower performance. I hope the authors can have more investigation and discussions on this point, which could be quite controversial given the common agreement that ML models usually benefit from bigger data. Thinking about this, I feel it may depend on different scenarios, such as different types of models used with different capacities to handle large data, and how you train and evaluate the model - the model with more input data may not get optimized which leads to underfitting. Taking one example, for experiments EX1-EX7, the optimized hyperparameters can be different with varying training data availability, and a fair comparison should be built on the optimized conditions of different models.

Response: The results presented here are for the pre-trained ML models in the entirely new study domain (the target). Therefore, our assertion does not conflict with the common agreement that ML models perform better with more training datasets (Figure 4). Our message is that using models forced by the greatest possible number of datasets does not necessarily ensure that those models will perform well. The newly added datasets may even add noise to the models. Specifically, we see that including European catchments may not be a good idea for creating pre-trained models to make reasonable predictions at targets in Central Asia and South Africa. This is probably due to the characteristics of European catchments being so different from those at the target and the fact that learned ML models from these catchments are not beneficial.

[10] I didn't understand the results shown in Figure 3 well. Are these the results on source (gauged) or target (ungauged) basins? Are they reported on the testing data, and if so how did you divide the testing data?

Response: In Figure 3, the results show the testing data for the sources from experiment EX7 (input datasets include S1 – North America, S2 – South America, and S3 – Western Europe). We did not fix the testing data; instead, we randomly divided it into training, cross-validation, and testing sets. One catchment could fall into a training set in one simulation but into a cross-validation or testing set in another simulation. Thus, Figure 3 shows the performance of models trained with different input datasets from EX7. From these results, we gained insight into the uncertainties of ML algorithms as they respond to different input datasets. More importantly, this method allowed us to apply a set of hyper-parameters (associated with different input datasets) from the source (EX7) to the target region. We will revise the manuscripts to help readers better understand our deliverable message in this figure.

**Conclusion:**

[11] As mentioned in the above comments, I feel the two key points in line 341 and line 343 are kind of contradictory regarding whether more diverse data can lead to better performance or not. The authors should carefully investigate this point before drawing a conclusion here. In addition, as mentioned previously, more analyses on the absolute PUR performance are needed to get the strong conclusion in line 351 that

these models can be capable of solving PUR problems in ungauged regions, especially given the deteriorated performance shown in Figure 8.

Response: We appreciate this comment. In line 341, we discuss the case that unseen data are in similar geographic region as more training data points likely to improve the prediction capacity of the pre-trained model at the same region.

In line 343, we aim at the performances of pre-trained models at an entirely new geographic location (i.e., transfering the model to new region). In this context, the common agreement that pre-trained models with more traning samples will peformed the best is not likely true. In line 351, we agree that the conclusion sounds too optimistic about the performance of the pre-trained models. Specifically, the pre-trained models have performed well in predicting streamflow in several months - but not all months. In our revision, we will revise this paragraph substantially to communicate our findings better and ensure that (any) caveats are presented in a transparent manner.

---

## Author Comment (AC3)

**Response to RC#3**

[1] I found this manuscript is very confusing. I am not sure about their numerical experiments. Before I have a good understanding of their experiments I cannot give a good review on the results. Below are my comments for now. I am happy to give more detailed review after I have a better understanding of their numerical experiments from their revised manuscirpt.

Response: Thank you for your comments. We have answered your concerns point by point below, with the hope that you can better understand our intent.

[2] The title mentioned "quantifying uncertainties in geographic extrapolation". I am wondering how the authors quantified the uncertainties. This uncertainty quantification is one of the objectives of this study if I understand the authors correctly, but I did not see any related discussion in the introduction till the results analysis.

Response: Our aim was to vary input datasets used for forcing in order to quantify uncertainties in our ML algorithms in a geographic extrapolation topic. The ML algorithm had different hyper-parameters at the source region when we forced it with different input datasets. Variations in the hyper-parameters resulted in different streamflow predictions at the target based on the pre-trained ML algorithm at the source. The overall best ML algorithm in our experiment was obtained from the algorithm (which, in our experiment, was eXtreme Gradient Boosting) having the most accurate predicted range for the streamflow at the target region. We will enhance the introduction to highlight the need for uncertainty quantification in ML algorithms.

[3] The conclusion in the abstract said "This study provides insight into the selection of input datasets and ML algorithms with different sets of hyperparameters for a geographic streamflow extrapolation." I am wondering what the insights are specifically.

Response: There are two items discussed here. Firstly, we trained ML algorithms with different input permutations to determine the impact of input sets on the capability of ML algorithms in an entirely new prediction domain having an unknown input-output relationship. Secondly, since variation of input datasets can result in different hyper-parameters for ML algorithms, we examined how such variations in hyper-parameters impact the predicted streamflow range in the new study domain.

[4] The effectiveness of transfer learning depends on the similarity of the source and the target. I am wondering whether the authors performed a similarity analysis which I think it is important to analyze the effectiveness of the extrapolation. And it might explain that adding more sample data from the sources did not improve the performance in predicting the targets.

Figure S1 shows the spatial pattern analysis using Uniform Manifold Approximation and Projection (UMAP), which untangles the inputs for source and target regions for twelve different months. It is interesting to note that the target catchments (rectangles) are primarily within catchments from the source, demonstrating the possibility that pre-trained ML over the source regions (circles, crosses and plus marks) can predict the output at the target regions.

[Figure]

Figure S1. Uniform Manifold Approximation and Projection (UMAP) clustering of the source and target regions (colored and symbolized by catchment). Source Region 1 (S1)-North America, Source Region 2 (S2)-South America, Source Region 3 (S3)-Europe; Target Region 1 (T1)-South Africa, Target Region 2 (T2)-Central Asia

[5] Line 107, what "hypothesis"?

Response: The learned/pre-trained ML algorithm at the source may be able to predict streamflow in the entirely new study domain (the target).

[6] Why specifically chose these three ML methods? How about the more recently widely used LSTM network? It is known that these three chosen ML methods cannot learn the temporal dependence and the memory effects of the dynamic inputs on streamflow outputs.

Response: This study focuses on a spatial prediction model rather than a temporal model. That is why we do not see the fit of LSTM. In addition, our sample dataset is not too large to train LSTM. The sample size used to train the model in our experiment can be a relatively small number of samples (please see Figure 4), while LSTM may not perform well even with multiple decades of records (daily time step) (Kratzert et al., 2019).

Reference:

Kratzert, F., Klotz, D., Shalev, G., Klambauer, G., Hochreiter, S., & Nearing, G. (2019). Towards learning universal, regional, and local hydrological behaviors via machine learning applied to large-sample datasets. Hydrology and Earth System Sciences, 23(12), 5089-5110.

[7] Did the authors consider the influence of lagged P and T on current streamflow when they designed the numerical simulations?

Response: We did not consider lagged time in our experiment. First, our experiment is not a time-series model but rather a spatial predictive model. In addition, we aggregated all variables to a climatological monthly time scale. With regard to results, lagged time analysis would not make sense.

[8] Please be specific about the input and output data. Both spatial and temporal data were considered, how the authors split the data for training-validation-testing in terms of both space (i.e., catchments) and time period. The description of 25%-25%-50% of the total number of data is very vague. I do not know what the total number of data represent?

Response: Each model has 16 input variables, including two dynamic variables (precipitation and air temperature) and 14 static/ invariant variables. Since our work was based on a climatological monthly timescale, we primarily focus on spatial scale but not temporal scale. For a typical experiment, the associated dataset was divided randomly by a ratio of 0.25, 0.25, and 0.5 for training, cross-validation and testing. We created 100 folds of these training, cross-validation, and testing datasets for each experiment. Table S1 below represents data characteristics for the first six input datasets (of 100 in total) from experiment 01 (EX01) and experiment 07 (EX07), respectively.

**Table S1**. Statistical description of input datasets for EX01 and EX07. The labels 'train', 'cv', and 'test' denote training, cross-validation, and testing datasets. The labels 'n', 'min', 'mean', and 'max' denote total sample size, minimum, mean, and maximum values.

| no | train.n | train.min | train.mean | train.max | cv.n | cv.min | cv.mean | cv.max | test.n | test.min | test.mean | test.max |
|---|---|---|---|---|---|---|---|---|---|---|---|---|
| **EX01** | | | | | | | | | | | | |
| 1 | 247 | 0.14 | 19.02 | 160.12 | 247 | 0.17 | 19.63 | 266.12 | 493 | 0.16 | 17.08 | 285.83 |
| 2 | 247 | 0.17 | 18.17 | 221.46 | 247 | 0.17 | 18.92 | 187.12 | 493 | 0.14 | 17.86 | 285.83 |
| 3 | 247 | 0.17 | 16.50 | 108.78 | 247 | 0.15 | 21.43 | 285.83 | 493 | 0.14 | 17.44 | 187.12 |
| 4 | 247 | 0.14 | 17.45 | 285.83 | 247 | 0.16 | 18.70 | 221.46 | 493 | 0.17 | 18.32 | 266.12 |
| 5 | 247 | 0.14 | 18.62 | 143.46 | 247 | 0.17 | 15.90 | 187.12 | 493 | 0.15 | 19.15 | 285.83 |
| 6 | 247 | 0.15 | 17.99 | 119.21 | 247 | 0.17 | 18.06 | 285.83 | 493 | 0.14 | 18.38 | 187.12 |
| 7 | 247 | 0.17 | 18.14 | 285.83 | 247 | 0.14 | 16.73 | 108.78 | 493 | 0.15 | 18.97 | 266.12 |
| **EX07** | | | | | | | | | | | | |
| 1 | 564 | 0.14 | 34.90 | 372.5 | 564 | 0.16 | 38.61 | 337.41 | 1129 | 0.17 | 35.66 | 338.39 |
| 2 | 564 | 0.15 | 35.46 | 372.5 | 564 | 0.17 | 37.88 | 337.41 | 1129 | 0.14 | 35.74 | 316.24 |
| 3 | 564 | 0.16 | 34.66 | 372.5 | 564 | 0.23 | 34.87 | 301.52 | 1129 | 0.14 | 37.65 | 338.39 |
| 4 | 564 | 0.22 | 36.77 | 298.2 | 564 | 0.16 | 36.54 | 372.5 | 1129 | 0.14 | 35.75 | 338.39 |
| 5 | 564 | 0.16 | 35.13 | 372.5 | 564 | 0.15 | 34.57 | 337.41 | 1129 | 0.14 | 37.56 | 316.24 |
| 6 | 564 | 0.23 | 35.50 | 266.32 | 564 | 0.16 | 40.43 | 372.5 | 1129 | 0.14 | 34.45 | 337.41 |

[9] I am confused about the local-based models. It said "using target catchments to train the ML algorithms", did it also include the source catchments or just target catchments?

Response: Local-based models only consider target catchments. We used these local-based models as benchmark results (our usual way of developing models) to examine the performance of the source-based models at the target.

[10] Figure 2. I am confused about the total data, i.e., training is about 25% of total. Did this total data include all five regions (source +target) or just source/target?

Response: The total data depend on the experiment; please refer to Table S1 in [8] for more detail. It should be noted that no experiment includes all five regions. This experiment was designed with the idea of developing pre-trained models from different source regions (North America, South America, and Western Europe). Uncertainties were analyzed after we used these pre-trained models to predict streamflow at the target region (Central Asia, South Africa).

[11] Table 3 and the 7 experiments need more explanation. I am not sure what these 7 experiments are.

Response: We designed different input imputations to get different pre-trained ML models (different hyper-parameters) at the source. Then we used these pre-trained ML models to predict climatological streamflow at the target. We will enhance our explanation for Table 3 in the revised manuscript.

[12] Line 241, for each of these 100 simulations, the hyperparameter tuning was performed and the best results were presented? Please clarify.

Response: Each simulation was forced by a different input set. Hyper-parameter tuning was performed and the metrics for the testing sets (unseen sets which existed during the hyper-parameter tuning process) were calculated. We will revise this sentence for clarification.